# Real-time measurement of spatial distance to external breakage hazards of transmission pole tower based on monocular vision

**Ruchao Liao**\*, **Duanjiao Li, Changyu Li, Wenxing Sun, Gao Liu, Cong Wang**

Guangdong Power Grid Co., Ltd., Guangzhou, China

\* A85746276@163.com

## Abstract

As the global economy continues to expand and energy demand increases, the size of power transmission networks continues to grow, making the safety monitoring of transmission towers increasingly important. To address the accuracy deficiencies of existing technologies in predicting external damage risks to transmission towers, this study proposes a real-time spatial distance measurement method based on monocular vision. The method first uses a Transformer network to optimize the distribution of pseudo point clouds and designs a 3D monocular vision distance measurement method based on LiDAR. Through validation on the KITTI 3D object detection dataset, the method achieved an average detection accuracy increase of 10.71% in easy scenarios and 2.18% to 7.85% in difficult scenarios compared to other methods. In addition, this study introduced a foreground target depth optimization method based on a 2D target detector and geometric constraints, which further improved the accuracy of 3D target detection. The innovation of the study is the optimization of the pseudo point cloud distribution using the transformer network, which effectively captured the global dependencies and improved the global consistency and local detail accuracy of the pseudo point clouds. The method proposed in the study provides a new approach for intelligent detection and recognition of power transmission lines, and provides a positive impetus for the fields of power engineering and computer vision.

## 1. Introduction

With the development of the global economy and the increase in the demand for energy consumption, the scale of the power transmission network is constantly expanding. The emergence of intelligent and automation technology promotes the development of smart grid, and promotes the efficiency improvement and intelligent detection of power transmission technology [1,2]. Power transmission towers are key components in power transmission lines. It is of positive research significance to

**Data availability statement:** All relevant data are within the manuscript and its Supporting Information files.

**Funding:** The author(s) received no specific funding for this work.

**Competing interests:** The authors have declared that no competing interests exist.

carry out real-time monitoring of transmission towers to reduce their hidden safety risks and hidden dangers, and to improve the operational safety of power lines (PLs) [3,4]. 3-dimension (3D) laser point cloud (PC) technology, as an advanced laser measurement technology, captures the spatial information on the surface of an object through the principle of laser range finder. However, the current 3D laser PC technology mostly relies on extracting the suspension points of transmission towers from the point cloud data (PCD) in the external breakage hidden danger monitoring of transmission towers. The distribution of pseudo point cloud (PPC) at the edge of the transmission tower has a large error with the distribution of the internal area to be determined. How to improve the data feature extraction of 3D laser PC technology to improve the efficiency of real-time measurement of transmission towers and other targets has become a current research hot spot.

C. Zong and Z. Wan proposed a unit rail segmentation method with an improved 3D PC segmentation model in an attempt to enhance the installation accuracy of unit rails on container ships. The method solved the errors during container loading due to low loading accuracy, welding shrinkage and expansion deviations by fitting the structure of unit rails on container ships [5]. L. Hui et al. developed an efficient PC learning network in order to solve the problem of excessive computational resource consumption of existing deep learning based methods. The method aggregated the local geometric features of the PC by the proposed lightweight ProxyConv neural network module to obtain a global descriptor of the PC for location identification [6]. A filtering method incorporating an anisotropic point error model and a simplified process was proposed by M. Ozendi et al. to address the problem of low-quality points in PCD acquired by light detection and ranging (LiDAR). The method effectively eliminated redundant data due to different positional quality patterns of the target object over multiple scans by taking into account the main error sources [7]. T. Y. Tang et al. addressed the limitations of using top-view images for LiDAR localization by proposing a method that utilized a public top-view image as a map proxy. The method simulated the acquisition of PCs scanned by LiDAR sensors located near the center of the top view image by converting the top view image into a collection of 2-dimensional (2D) cloud points, thus weakening the large domain discrepancy between the LiDAR data and the top view image [8]. Z. Zhang et al. proposed a perception algorithm inspired by the eagle's eye. The method achieved adaptive high-dynamic-range stabilization by simulating the physiological structures of "deep concavity" and "shallow concavity" in the eagle's eye, thus obtaining a 78.6% improvement in LiDAR PC bias in fixed-distance measurements [9]. Aiming at the efficiency of LiDAR-based simultaneous localization and mapping (SLAM) in the field of robot mapping, X. Yue analyzed the application of different types and configurations of LiDAR in SLAM system. It was found that multi-robot collaborative mapping and multi-source fusion SLAM system based on 3D LiDAR, combined with deep learning, could improve the accuracy of robot mapping [10].

With its accurate 3D data method that can capture intricate settings or structures, 3D laser PC technology has found extensive application in a variety of areas, including electric power, agriculture, and medical. The use of airborne or ground-based

LiDAR for transmission line inspection can achieve high-precision 3D modeling of the line, and timely and effective access to defects or safety hazards of transmission lines [11,12]. X. Liu et al. proposed a point-by-point multi-layer perceptron (MLP)-based semantic segmentation (SS) network to address the SS of key objects in PL inspection and the low efficiency in dealing with a large amount of PCD and missing PCs of PL. The method effectively solved the point missing problem by designing a local coding module to improve the segmentation ability of the network in the corridor [13]. A framework for automated vegetation monitoring along PL was proposed by M. Gazzea et al. in response to the drawbacks of slower and less effective vegetation management and environmental inspections. The method reduced the cost and time of PL monitoring by utilizing satellite data analysis as an alternative to ground patrols and inspections by helicopters or drones [14]. D. Shokri et al. proposed an intelligent algorithm combining preprocessing, pole extraction and cable extraction to address the low efficiency of monitoring, maintenance and organization of PL corridors. The method detected the search area containing the lines by automatically extracting the poles and cables data from a moving ground LiDAR PC using a Hough transform algorithm. This achieved 100% average correctness and 97% completeness extraction for utility poles [15]. A. Al Najjar et al. proposed a two-stage method for detecting vegetation encroachment on PLs in urban areas, combining laser point cloud technology and point convolutional neural networks. By slicing the map and selecting informative parts, as well as conducting proximity analysis between vegetation and PL voxels, precise and automatic detection of vegetation encroachment on urban PLs was achieved, thereby optimizing vegetation management and active maintenance in urban environments [16]. M. García-Fernández et al. proposed a method for comparing different scanning strategies to address the problem of difficulty in improving the scanning throughput of multi-channel ground penetrating radar synthetic aperture radar systems carried by drones. By considering different values of inter track spacing to generate dense and sparse sampling distributions, detection efficiency could be improved while maintaining image quality [17]. C. Li et al. put forward an accurate parallel laser line scanning system to detect the depth of structural defects on concrete surface. Through the triangulation device composed of digital camera, double-line laser diode and positioning rigid arm, the image processing algorithm for extracting depth information from distorted laser strips was realized. It improved the accuracy and efficiency of evaluating the depth of defects at different distances [18]. M. M. Hosseini et al. addressed the inefficient and time-consuming problem of manual operations in the process of power tower damage assessment by proposing an automated model that used an unmanned aerial vehicle (UAV) to capture images and transmit them in real time to an intelligent damage classification and assessment unit. The model's integration of four convolutional neural networks facilitated the learning of the tower condition from images, the extraction of image features, and the training of automated intelligent tools to replace manual fault location and damage assessment [19]. Chen et al. proposed a comprehensive risk assessment framework based on transmission tower geometry and topography to address the problem that transmission towers are prone to collapsing and triggering large-scale power outages and electrocution risks when subjected to heavy rainfall and flooding. A series of new point cloud segmentation and fitting algorithms are developed to accurately estimate the tilt angle of transmission towers by utilizing 3D point cloud data acquired from aerial LiDAR [20].

Combined with the above, it can be observed that scholars at home and abroad have conducted various researches on PPC distribution optimization, and have achieved good research results in many areas. However, there is a PPC long-tailed distribution effect in the process of point cloud image acquisition using LiDAR technology for transmission tower external damage risk detection. In the existing transmission tower external damage risk prediction technology, the processing time of the traditional LiDAR technology is about 3-5s, and the data processing and analysis is too time-consuming to meet the real-time demand. In addition, the completeness of the feature extraction of the towers is only about 97%, and there is a 3% error between the distribution of PPC at the edges of the towers and the distribution of the internal area to be determined, which limits the accuracy of the prediction model. And there are still some research gaps on the limitations of stereo vision in long-distance measurement. Therefore, the study proposes a real-time spatial distance measurement method for transmission tower external breakage hidden danger based on monocular vision with a view to improve the efficiency of predicting hidden danger on transmission towers. The distribution of PPC is optimized by

using Transformer network as an encoder, and a distance measurement method for monocular vision is further designed. The novelty of the study is that the encoder-decoder structure is used for PC feature distribution enhancement. The study also achieves adequate extraction and pseudo point cloud distribution optimization (PPCDO) of the actual PCD in a way that enhances the feature information of the PCD.

The study proposes a real-time measurement method for the spatial distance of external breakage and hidden dangers of transmission towers based on monocular vision. This method employs the Transformer network to optimize the distribution of PPC and combines the depth optimization of foreground targets to enhance the accuracy and efficiency of 3D target detection. By optimizing the PPC with the Transformer encoder, the CD error is effectively reduced, thereby improving the accuracy of the PCD. Based on the optimized PPC, a 3D monocular visual distance measurement method based on LiDAR is developed in this study, which further improves the measurement accuracy. The experimental validation results confirm the superiority and effectiveness of the proposed method, and demonstrate its practical application value on the inspection data of Guangdong Power Grid Company. The real-time measurement method proposed by the study offers a novel approach for intelligent detection and recognition of power transmission lines. Furthermore, the PPCDO strategy is of considerable significance for enhancing image measurement and recognition.

The overall structure of the study consists of three sections. In the first section, the real-time measurement method of the external breakage hidden danger spatial distance of transmission towers based on monocular vision is studied and designed. In the second section, the proposed method is experimented and analyzed. The third section summarizes the experimental results and indicates the future research direction.

## 2. Methods and materials

To improve the efficiency of detecting external breakage hidden danger for transmission towers, the study firstly optimizes the design of PPC distribution on the basis of Transformer network. Second, a foreground objective depth optimization method is proposed. On this basis, a real-time spatial distance measurement method based on monocular vision is further designed by combining PPCDO and foreground target depth optimization.

### 2.1. Transformer-based pseudo point cloud distribution optimization

To adjust the large error between the distribution of PPC at the edge of the target object and the distribution characteristics of a specific region inside the target, the study first defines the target problem of how to make the simulated PC closer to the actual PC acquired by LiDAR as the PC distribution transformation problem. The main factor influencing the discrepancy between the distribution of the real PC and the simulated PC is the blurring of the deep location data of the PC during the acquisition phase. Therefore, the study sets the distribution information of the PC as the position information in the 3D spatial coordinate system, and utilizes the bilinear image processing method to obtain the quantitative information of the PC [21,22]. Fig 1 depicts the unique optimization procedure.

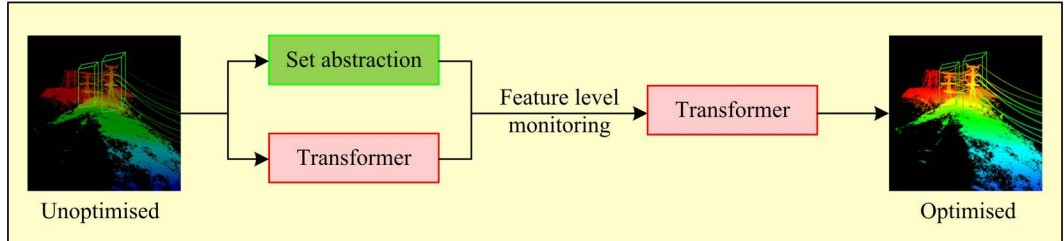

**Fig 1. Transformer-based pseudo-point cloud distribution optimization process.**

In Fig 1, while performing the feature extraction process for different PCs, the study utilizes PointNet++ to monitor the whole process. The PointNet++ network solves some of the limitations of the original PointNet in dealing with PCD with complex structures, especially in extracting local features and multi-scale features, through an innovative network structure [23,24]. PointNet learns global features by directly processing the entire point cloud. It considers each point in the PC as an independent sample, ignoring the spatial relationship and local structure information between points. Point-Net++ introduces a layered architecture that uses different scales to capture the local structure in the PC. This is achieved by sampling and grouping PCs at different scales, allowing the network to learn the characteristics of local areas. In addition, PointNet++ can capture local features of different scales by setting up multiple layers, each with different receptive fields. Due to the lack of local feature capture, PointNet has limited generalization ability, especially when dealing with PCD with changing shapes. PointNet++, on the other hand, by introducing local feature capture, multi-scale feature fusion and hierarchical architecture, is able to improve its ability to handle complex PCDs, enabling it to perform well in a variety of PC-related tasks [25]. Therefore, the integration of global information can be enhanced by using it to encode PC properties. The specific network structure is shown in Fig 2.

In Fig 2, the PointNet++ network has a point set abstraction module that extracts key features and structural information from raw PCD. The module consists of three main layers: sampling, integration, and PointNet. Among them, the sampling layer mainly performs sampling of the whole received PCDset according to the farthest point sampling (FPS). Moreover, one point is randomly selected as the first center point, and iteration is performed to obtain multiple center points. FPS aims to select a uniform set of points from a PC for efficient processing and analysis. The method ensures sampling uniformity by selecting the points that are furthest away from the current set of points. This helps the network capture the global structure of

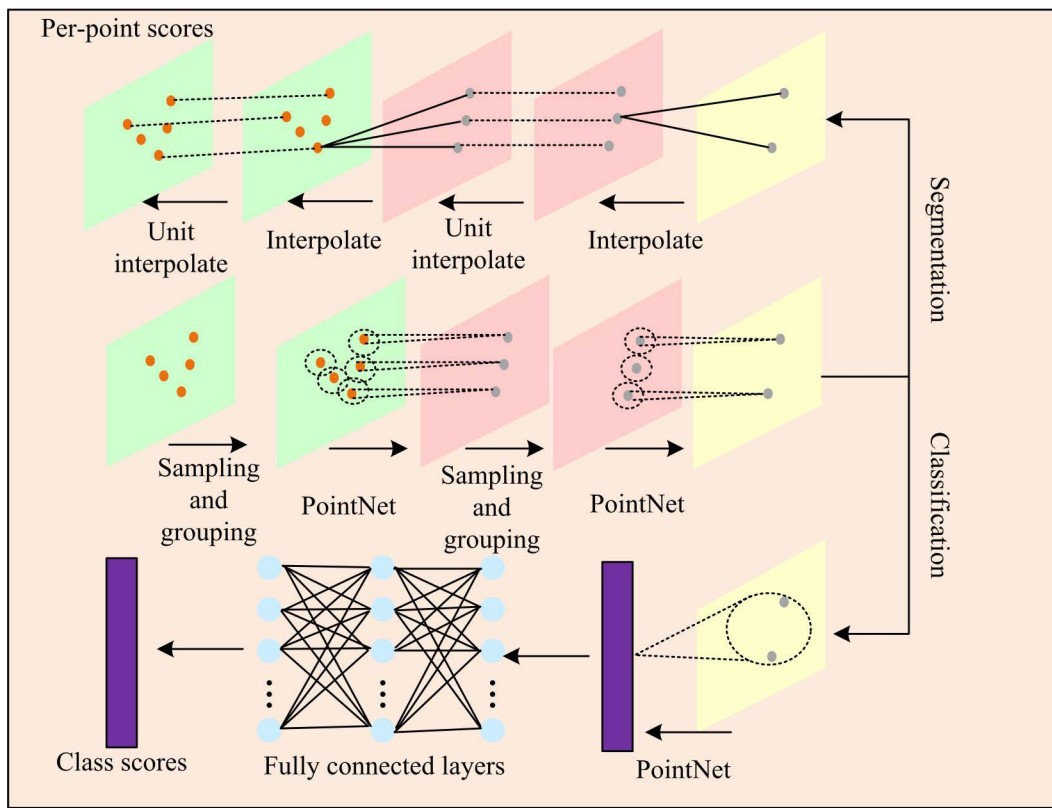

**Fig 2. PointNet++ network framework.**

the PCs and reduces the amount of computation required for large-scale data processing. In short, FPS improves the efficiency and effectiveness of PCD processing through uniform sampling. Random selection of the first centroid of the FPS algorithm is essential to ensure unbiased and uniform PC sampling. This selection method circumvents any bias towards particular regions, bolsters the resilience of the algorithm, and facilitates the comprehensive encapsulation of the global structure of the PCs, thereby furnishing a homogeneously dispersed set of points for subsequent feature extraction and analysis.

The objective of the integration layer is to merge the local PC features extracted from the sampling layer and construct a unified feature representation reflecting the global structure of the PCD. This is achieved through the attention mechanism, which provides comprehensive data support for subsequent analyses. This process employs spherical queries to delineate local regions and integrate these local features to capture multi-scale information. The PointNet layer, on the other hand, is mainly responsible for extracting features from the resulting PC collection and using it as input for subsequent tasks.

As one of the core components of PointNet++, the Set Abstraction module is responsible for extracting local features from the original PC. The module first uses FPS to select a set of scattered points as a "set". For each center point, a local region is defined around it using Ball Query, and all points within this region are extracted from the original PC. PointNet++ realizes multi-scale feature extraction through multiple set abstraction modules, each module can have different radius and sampling density, thus capturing local features of different scales. After multi-scale local feature extraction, PointNet++ aggregates all local features through a global pooling layer to generate global feature vectors (FVs). This global FV contains the geometric and semantic information of the entire PC.

The study performs encoder construction of PC features based on PointNet++. The auto-correlation processing unit of the Transformer network is then utilized to reintegrate the encoder-extracted multi-resolution PC descriptions for auto-correlation feature extraction. In this case, the PC FVs are pre-calculated with tri-linear layer mapping before the attention calculation. The specific calculation formula is shown in Equation (1) [26,27].

$$q, k, v = L(F_{\Sigma}^{N_i})$$
(1)

In Equation (1), $q$ is the query vector. $k$ is the key-in vector used to determine the attention score. $v$ denotes the value vector containing the actual data information. $L(F_{\Sigma}^{N_i})$ denotes the feed-forward neural network. $F_{\Sigma}^{N_i}$ denotes the original FV input to the MLP. $i$ is the layer level. $N$ is the total point sets obtained from network sampling. After the attention weighting process, the PC FV expression formula $S_a$ is shown in Equation (2).

$$S_a = soft\max\left(\frac{qk^T}{\sqrt{d_i}}\right)v$$
(2)

In Equation (2), $soft\max$ denotes the Softmax function, it is mainly used to normalize the attention scores, ensuring that the weights of each FV sum to 1. $qk^T$ denotes the matrix multiplication that measures the degree of match between the query and the key. $\sqrt{d_i}$ denotes the scaling factor. $d_i$ denotes the dimension of the FV. Moreover, in Transformer network, the weighted feature expression formula obtained from PointNet++ output is shown in Equation (3).

$$f_{attn} = LayerNorm(f_k + S_a(f_k))$$
(3)

In Equation (3), $f_{attn}$ denotes the weighted features after processing through the self-attention layer. $LayerNorm$ denotes the normalized network layer. $f_k$ denotes the attributes or metrics of the data points input to the model for performing learning. Equation (3) is used to compute the weighted FVs after the self-attention layer processing, which is used to further optimize the global consistency of the point cloud features. The final output of Transformer's FV expression formula is shown in Equation (4).

$$f_{enh} = LayerNorm(f_{attn} + FeedForward(f_{attn}))$$
(4)

Equation (4), $f_{enh}$ denotes the FV of the final output of Transformer. *FeedForward* denotes the feed-forward network of Transformer network. Therefore, the flow of the proposed encoder combining PointNet++ feature extraction with Transformer based PPCDO for decoder is shown in Fig 3.

In Fig 3, both the encoder and decoder of PPCDO require multi-head self-attention (MHSA) and MLP implementation. Among them, MHSA ensures that the encoder and decoder process information in parallel at the same time. This improves the network's ability to understand the real PC feature data and thus improves the generalization ability of the network. MHSA facilitates parallel processing by dividing the self-attention calculation into multi-heads, with each head focusing on a distinct feature subspace of the input data. In the encoder, MHSA parallel processing enhances the efficiency of feature extraction, whereas in the decoder, it enables the model to examine a multitude of potential feature fusion pathways in parallel and to optimize the distribution of PCD. By introducing PCD at different scales and from different perspectives during training, MHSA enables the network to learn more robust feature representations. While MLP mainly performs classification of image PC features during encoding and decoding. In the encoding phase, the MLP employs its multi-layer structure to non-linearly transform the PCD, thereby extracting high-level features that encompass both local and global structural information. In the decoding phase, the MLP inverses the high-level features, returning them to the original space for accurate PC classification. The MLP achieves accurate classification of PCD by learning complex nonlinear mappings that transform the abstract feature space into concrete category labels.

Fig 3(a) shows the overall encoding process of the feature encoder. The semantic information of the corresponding level of features is deepened by repeatedly accumulating the basic constructs of multiple levels and connecting the levels at each network level of PointNet++. In Fig 3(b), the decoder of PPCDO mainly performs inference computation on a set of FVs at multiple levels to optimize the distribution of the PC. It contains the Transformer network with repeated accumulation of multiple levels, and the number of repeated accumulations of the basic Transformer constructs is the same as that of PointNet++. The decoder module of the transformer facilitates the restoration of the FV output by the encoder to the spatial distribution of the PC. The features are then fused at multiple scales through the use of MHSA, which optimizes the detailed information of the PC distribution. Ultimately, the error between the optimized PPC and the actual PC is calculated by the chamfer distance (CD) loss function, and the network parameters are adjusted by back-propagation. In this

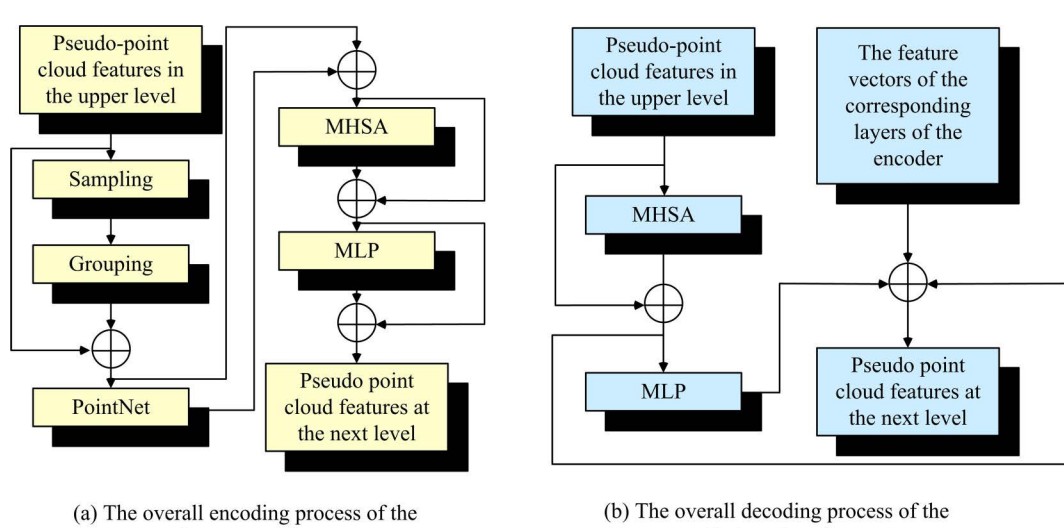

(a) The overall encoding process of the feature encoder

(b) The overall decoding process of the feature decoder

**Fig 3. Flow of feature encoder with pseudo point cloud distribution optimization decoder.**

case, the loss $Loss_s$ expression formula for the mean and variance between the FVs at the corresponding level between the decoder and the encoder is shown in Equation (5).

$$Loss_s = \left\| u(f_k^{dec}) - u(f_k^{enc}) \right\|_2 + \left\| \sigma(f_k^{dec}) - \sigma(f_k^{enc}) \right\|_2 \tag{5}$$

In Equation (5), $u(\cdot)$ denotes the mean value calculation on the set of FVs. $f_k^{dec}$ is the set of FVs of PPC after processing by a certain layer of encoder. $f_k^{enc}$ denotes the set of FVs of real PC after processing by a certain layer of encoder. $\sigma(\cdot)$ denotes the standard deviation computation of the set of FVs. The loss function calculation formula for PPC after final optimization is shown in Equation (6).

$$Loss_{CD} = \frac{1}{N} \left( \sum_{x \in P_{op}} \min_{y \in P_{gt}} \|x - y\|^2 + \sum_{y \in P_{gt}} \min_{x \in P_{op}} \|x - y\|^2 \right) \tag{6}$$

In Equation (6), $Loss_{CD}$ denotes the CD loss function. $CD$ denotes CD. $P_{op}$ denotes the optimized PPCDset. $x$ denotes a PC in $P_{op}$. $P_{gt}$ denotes real PCDset. $y$ denotes a PC in $P_{gt}$. $\|x - y\|^2$ is the squared Euclidean distance between $x$ and $y$. The overall loss $Loss$ calculation formula for PPC and real PC is shown in Equation (7).

$$Loss = \sum_m Loss_s^m + Loss_{CD} \tag{7}$$

In Equation (7), $m$ denotes the full number of feature layers of the PC in the network. The study utilizes adaptive supervision with multiple layers to modulate the designed decoder. To guarantee the convergence speed and effectiveness of the network training process, various supervised signal levels are utilized to enhance performance [28,29]. Meanwhile, high performance end-to-end network is constructed by comprehensively training the crosstalk between encoder and decoder. This increases target identification accuracy and allows PPC feature distribution to be optimized without the need for an actual PC.

## 2.2. Monocular vision measurement based on pseudo point cloud distribution optimization

Based on the previously proposed PPCDO method, the study further proceeds with the design of the transmission pole tower external breakage hidden danger monocular vision measurement method. Since the process of monocular vision predicting the depth information of the scene is not sensitive to the classification of foreground and background, the study proposes an optimization method focusing on the depth of foreground targets. A 2D target detector is used to replace the 2D instance segmentation and the network is trained using uncertainty learning (UL) [30]. The 2D target detector can effectively detect foreground targets in images with complex background situations and can clearly distinguish between foreground and background. The core idea of this stage is to extract the target region in the image by 2D target detection and estimate its depth using deep learning models [31]. Compared with the traditional method, the 2D instance segmentation method can deal with the target boundary more accurately, thus improving the depth optimisation of the foreground target. The study first uses the CenterMask network as the instance segmentation model for the proposed method in order to improve the joint training effect with UL. The specific framework is shown in Fig 4.

Fig 4 illustrates the structure of CenterMask network, which consists of spatial attention module (SAM) and feature channel attention module (FCAM). First, the input image is extracted from the feature image by the backbone network and fed into the feature pyramid network (FPN) to obtain the multi-scale feature image. Then, the image features are localized by the predictive bounding box module of the full convolutional single-stage target detection (FCOS). Finally, the image segmentation is completed by the spatial attention-guided mask (SAG-Mask). This

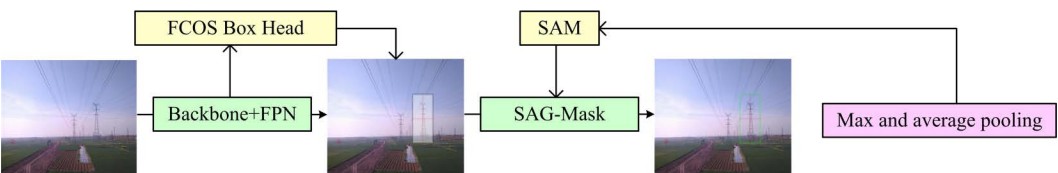

**Fig 4. CenterMask network.**

process not only effectively extracts the target features in the image, but also enhances the ability of the model to recognize the target region through the attention mechanism. After the CenterMask network has completed the instance segmentation, UL trains and learns the model. It primarily trains the model's judgment and detection skills, taking into account the size, kind, and pixel location of the image detection target, among other details. The particular procedure is depicted in Fig 5.

In Fig 5, the study first generates focus ranges based on each estimated obtained 2D target image. Moreover, the deep image, which is pre-estimated and measured, is integrated with the 2D instance cut image. On this basis, the foreground target region obtained from the depth image with the 2D instance cut is based on the depth image. The study introduces a geometrically constrained geometrically constrained depth refinement (GCDR) module to realize the UL learning of the model [32,33]. GCDR improves the stability of GCDR by refining the depths of close objects and optimizing the depth estimates of distant objects. For long-range targets, the GCDR module is able to optimise the depth estimation of the target and improve the depth accuracy of long-range targets. It is assumed that the 3D height prediction of each object is generated by adding a 3D height regression head to the CentreMask model, and the 3D height has Laplace distribution randomness [34]. By combining the 3D height prediction of the target and the depth regression head, the randomness of the Laplace distribution is exploited for the correction of depth estimation. Therefore, the probability density function $f(x)$ of the Laplace random variable is shown in Equation (8).

$$f(x) = \frac{1}{2\beta} e^{\frac{-|x-u|}{\beta}}$$

(8)

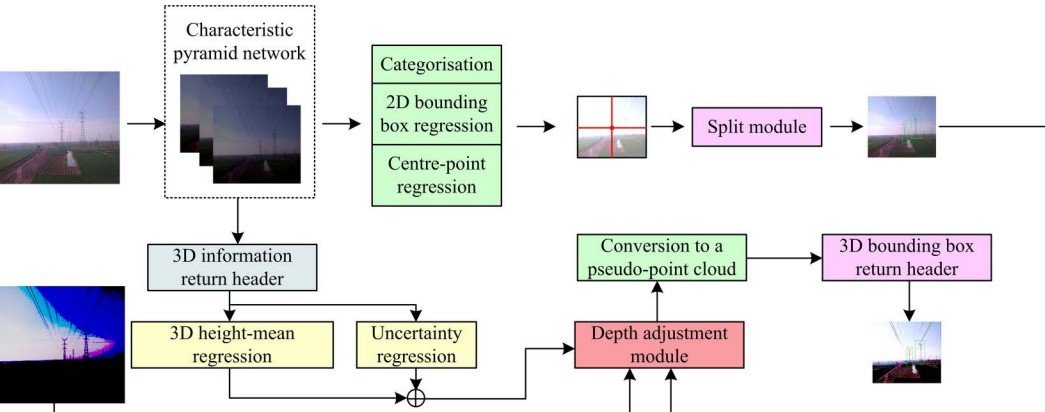

**Fig 5. Deep optimization process considering foreground goals.**

In Equation (8), $\beta$ is the distribution scale parameter. $e$ denotes the logarithm. Moreover, in order to detect the deep position information of the target at a long distance, the study introduces a distance sensitivity factor (DSF) as a constraint term penalty. The specific calculation formula is shown in Equation (9).

$$\begin{cases} \gamma = \log(d_g) \\ d_g = \dfrac{\sum\limits_{i=1}^{M} \left(d_{min} + \frac{d_{max} - d_{min}}{B(B+1)} \cdot i(i+1)\right)}{M} \end{cases} \tag{9}$$

In Equation (9), $\gamma$ denotes the DSF. $d_g$ denotes the average value of depth values in a single space at depth where the target is located. $M$ denotes the number of depth spaces. $d_{min}$ and $d_{max}$ denote the minimum and maximum spatial depth values, respectively. The formula for calculating the loss function $Loss_H$ for each 3D physical height obtained from model training is then shown in Equation (10).

$$Loss_H = \frac{\sqrt{2}}{\sigma_H} \left| u_H - H_{gt} \right| + \log(\gamma * \sigma_H) \tag{10}$$

In Equation (10), $H$ denotes the 3D height of the target. $\sigma_H$ denotes the randomness of the initial output result. $u_H$ denotes the initial output of the 3D regression head. $H_{gt}$ denotes the physical height of the object labeled with its true value. After obtaining the 3D spatial coordinate information of the target using the depth prediction module of CenterMask, the study further utilizes the spatial transformation function to calculate the depth spatial distribution $B_{pred}$ of the target. Equation (11) displays the particular calculating formula.

$$\begin{cases} B_{pred} = \frac{j \cdot H}{h} = \frac{j \cdot (\beta_H \chi + u_H)}{h} \\ u_{bias} = C_{mask} + \frac{1}{2} \end{cases} \tag{11}$$

In Equation (11), $j$ denotes the focal length of the camera lens. $h$ denotes the projection height of the target in the depth image. $\chi$ denotes the Laplace random variable. $\beta_H$ and $u_H$ denote the parameters of the Laplace distribution. $u_{bias}$ denotes the mean value of the redundant bias term generated by a pre-estimated depth. $C_{mask}$ denotes the average depth. The final predicted depth $D$ Laplace distribution of the target is calculated as shown in Equation (12).

$$D = La\left(\frac{j \cdot u_H}{h}, \frac{j \cdot \beta_H}{h}\right) + La(u_{bias}, \sigma_{bias}) \tag{12}$$

In Equation (12), $La(\bullet)$ denotes the Laplace distribution function. $\sigma_{bias}$ denotes the product of UL learning standard deviation and DSF. The final UL of the depth distribution mainly consists of the random metric of the projection and the random metric of the deviation obtained by learning. Therefore, the depth refinement loss function $Loss_{depth}$ of the GCDR module is updated as shown in Equation (13).

$$Loss_{depth} = \frac{\sqrt{2}}{\sqrt{\left(\frac{j \cdot \beta_H}{h}\right)^2 + \sigma_{bias}^2}} \left| \left(\frac{j \cdot u_H}{h} + u_{bias}\right) - d_{gt} \right| + \log\left(\gamma * \sqrt{\left(\frac{j \cdot \beta_H}{h}\right)^2 + \sigma_{bias}^2}\right) \tag{13}$$

In Equation (13), $d_{gt}$ denotes the actual depth value of the labeled target. Furthermore, the objective is to enhance the sensitivity to the distance and position information of the target in 3D space with respect to a reference point or reference plane. To this end, the natural exponential function is being investigated as a means of calculating the depth information of the target's distance in three-dimensional space, which is then converted to the confidence level of the coordinate system.

Moreover, the depth information within the target 2D segmentation is calculated. The specific calculation formula is shown in Equation (14).

$$\begin{cases} c_{depth} = \exp(-\sqrt{\left(\frac{j \cdot \beta_H}{h}\right)^2 + \sigma_{bias}^2}) \\ W_{final} = d_g \cdot (1 - c_{depth}) + D \cdot c_{depth} \end{cases} \tag{14}$$

In Equation (14), $c_{depth}$ denotes the depth formulation score. $\exp(\cdot)$ denotes the expectation function. $W_{final}$ denotes the final refined depth estimate. Therefore, combining the above, the flow of the study's proposed monocular vision-based method for real-time measurement of the spatial distance of transmission pole tower external breakage hidden danger is shown in Fig 6.

Fig 6 illustrates the complete process of the real-time measurement method for the spatial distance to external break hazards of transmission towers based on monocular vision. The method commences with the acquisition of the PCD of the transmission lines, subsequently followed by a series of pre-processing steps that include noise reduction and contrast enhancement. Concurrently, the optimized depth of the foreground targets is integrated with the original images to generate the PPCD. Secondly, based on the previously proposed Transformer, PCD optimization of transmission towers is performed using PPCDO and further PC key features such as wire edges, insulator locations, etc. are extracted. Moreover, the feature information is used for further hidden hazard analysis, such as identifying problems such as broken wires and damaged insulators. By extracting the visual features of the transmission tower from monocular images, candidate target regions are generated on the fused features. Moreover, it is projected into the depth map to further extract the PCs of the optic vertebrae.On this basis, the PCD is bis-classified by using PointNet or PointNet++, the 3D bounding box of the image detection target of the transmission towers is estimated. Furthermore, the candidate regions are used for the external breakage hidden danger of the transmission towers classification and identification. Finally, the transmission pole tower target detection and output are performed by point-voxel region convolutional neural networks (PVRCNN).

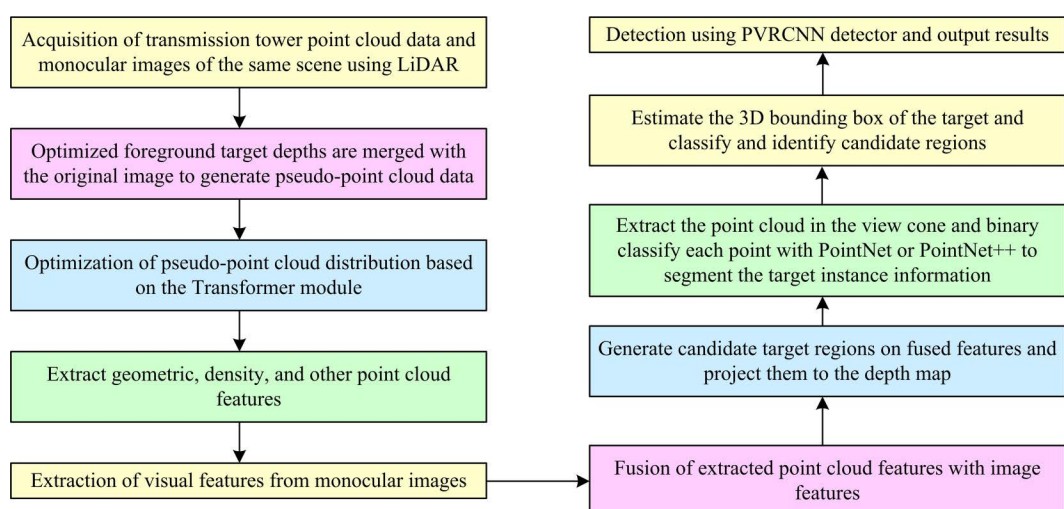

**Fig 6. Process of real-time spatial distance measurement method based on monocular vision for transmission tower external breakage hidden danger.**

## 3. Results

To validate the effectiveness of the proposed monocular vision based spatial distance measurement method for transmission tower external breakage hidden danger, the study firstly validates the Transformer based PPCDO method based on Transformer is validated. Second, the proposed monocular vision measurement method is validated and analyzed.

### 3.1. Experimental verification of relevant parameters

The study utilizes Karlsruhe Institute of Technology and Toyota Technological Institute at Chicago (KITTI) for experiments and analysis. The dataset includes a wide variety of sensor data such as high-resolution color and grayscale video, LIDAR scans, localization, and more. This is of great application in developing and evaluating stereo vision, 3D object detection and 3D tracking. Each image in the KITTI dataset is labeled with an accurate 3D bounding box, including information such as the category, location, size, and direction of the object. This labeling information provides accurate supervision signals for training and evaluation of the 3D target detection model. The dataset contains 14999 images. It is split up into three sets for the study: 3712 photos for training, 3769 images for validation, and 7518 images for testing. In addition, the dataset is split into three categories: simple, medium, and challenging, based on the picture recognition complexity, occlusion range, and pixel value. The specific definitions are shown in Fig 7.

In Fig 7, the study categorizes the KITTI dataset into three validation levels of easy, medium, and difficult based on the target image height, visibility, and target truncation, etc. At the same time, the research uses Jaccard index (intersection over union, IoU), CD, average precision (AP), precision rate and recall rate as verification indicators, and verifies and evaluates the performance of the proposed method. All images are normalized prior to input into the model to reduce the influence of illumination changes. Specifically, each pixel value of the image is normalized to the range [0,1]. The PCD obtained by LiDAR scanning is first converted into points in the 3D coordinate system, and then the number of points is reduced by voxelization for subsequent processing. The pre-processing steps are mainly realized by Python language and OpenCV library. The processing of PC depends on PC library.

The model's PointNet++ uses two Set Abstraction layers, and each layer contains 512 points. Transformer: Six Transformer blocks are used, and each block contains eight attention heads. Convolution layer: 4 convolution layers are used, and each convolution layer has 32 3x3 filters. Fully connected layer: Two fully connected layers are used. Moreover, the first fully connected layer has 128 neurons and the second fully connected layer has 64 neurons. The experimental hardware environment is GeForce RTX 3090 GPU, and the memory size is 48 GB. The CPU is Intel Core i9 and the RAM is 512GB. The operating system is Ubuntu 20.04 LTS and the programming language is Python 3.8.

### 3.2. Validation of transformer-based ppcdo approach

To verify the validity of PPC distribution on the basis of Transformer network, PointNet++ is firstly utilized as the main basic network for data feature acquisition. Moreover, Transformer is added to the feature extraction unit of each PC collection for image global feature information acquisition. Meanwhile, the study utilizes the validation set of KITTI dataset for experiments. The encoder is trained on the training set of the KITTI data set for 100 epochs. The cosine annealing algorithm is used as

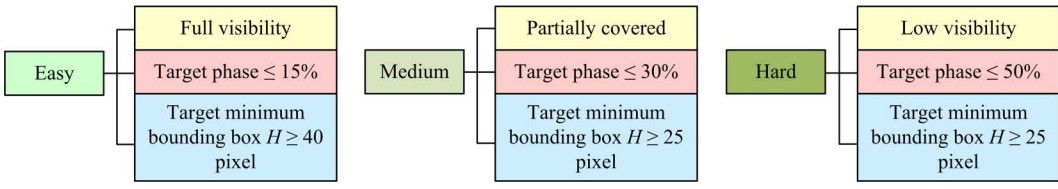

**Fig 7. KITTI dataset classification results.**

the model training strategy. The initial learning rate is set to 0.0001, and the learning rate of every 10 epochs is attenuated to the original 0.1. Adam: The default parameters is used, β1＝0.9 and β2＝0.999. Firstly, the training efficiency and loss curve is compared between using Transformer module to optimize PPC distribution and not optimizing it. As shown in Fig 8.

Fig 8(a) shows the change curve of model training loss before and after optimization. It can be seen that as the number of iterations increases, the loss values of both models keep decreasing and gradually converge. When the number of iterations is about 60, the loss curve of the Transforme model converges. The Transformer optimised model proposed in this study does not show overfitting and its loss value is lower than the unoptimised model after 10 iterations. This shows that the optimization strategy is effective and reliable. Concurrently, the convergence velocity of the proposed model is less rapid than that of the non-optimized model. This discrepancy may be attributed to the fact that the non-improved model is trained using pre-trained weights, whereas the proposed model is primarily trained through comprehensive retraining. Comparing the AP values of the two models in Fig 8(b), it can be concluded that the optimized Transformer model has better AP values than the unimproved model. Second, the CD error between PPC and PC is compared between using Transformer module to optimize PPC distribution and not optimizing it, as shown in Table 1.

In Table 1, when the IoU threshold in 3D target detection is 0.7, the AP obtained from the model after optimizing the PPC distribution using the Transformer module is significantly better than the un-optimized results. The output results under bird's eye view (BEV) also demonstrate the performance of PPCDO based on Transformer. In contrast, comparing the CD errors of the two methods before and after applying Transformer, it can be observed that the CD error between PPC and real PC decreases by 32.23% after optimization using Transformer. This indicates that the optimization of PPC distribution using Transformer network can improve the correlation of different points in the corona and reduce the gap between PPC and real PC. At the same time, the data indicates that the degree of correspondence between the optimized PPC and the actual PC is greater, indicating that each point in the PC is closer to its corresponding point in the target PC. The lower CD error indicates that the PC generation process preserves more detailed information, including edges, corners, and surface textures. Based on this, the study goes on to present the F-PointNet target detector for comparison with the proposed optimization approach. The particular outcomes are displayed in Fig 9.

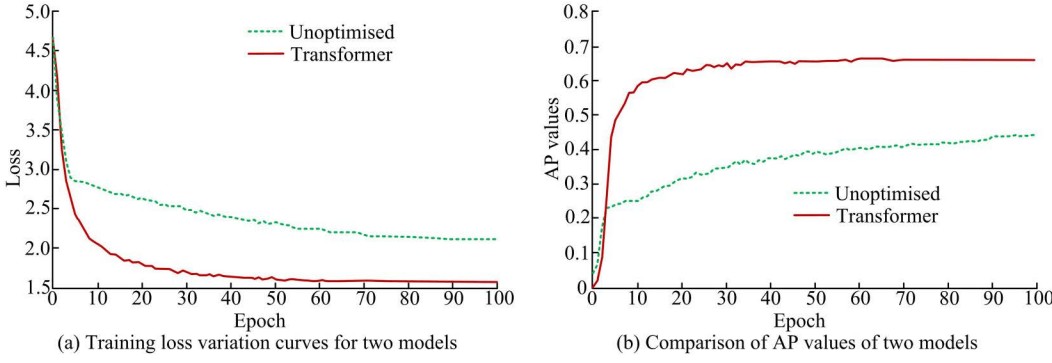

(a) Training loss variation curves for two models (b) Comparison of AP values of two models

**Fig 8. Comparison of training results between two methods.**

**Table 1. Performance comparison before and after applying transformer module optimization.**

| Method | CD(×104) | IoU3D = 0.7 (%) | | | IoUBEV＝0.7 (%) | | |
|---|---|---|---|---|---|---|---|
| | | Easy | Medium | Hard | Easy | Medium | Hard |
| Un-optimized | 15.36 | 42.11 | 25.48 | 21.86 | 53.12 | 32.57 | 28.77 |
| Transformer | 10.41 | 44.63 | 26.74 | 22.64 | 56.65 | 33.43 | 29.45 |

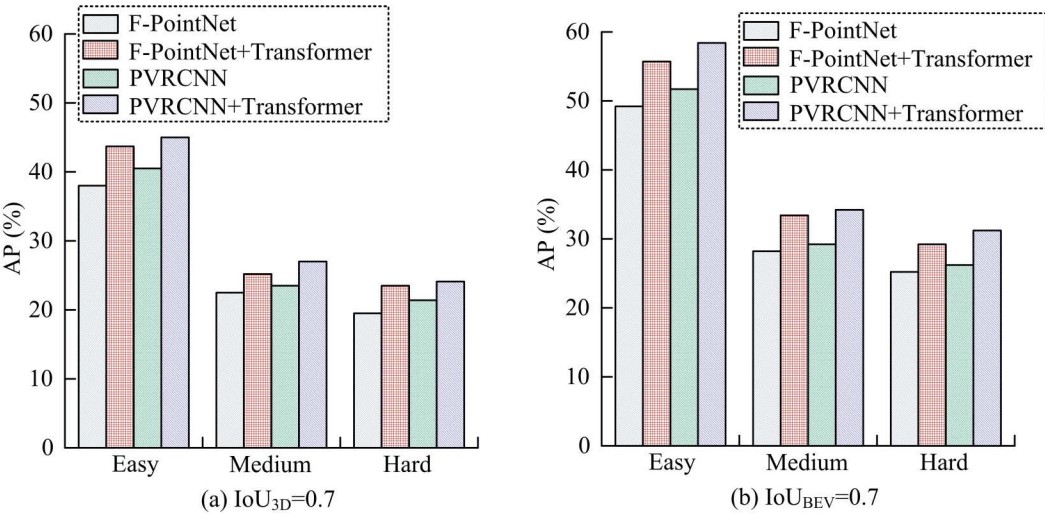

**Fig 9. Performance validation of the proposed method with different 3D target detectors.**

Fig 9(a) The validation results under IoU3D = 0.7 show that after the optimization of PPC using Transformer, the outputs of both target detectors, F-PointNet and PVRCNN, are significantly better than the un-optimized results. However, the proposed PVRCNN+Transformer method of the study increases the AP values by 1.35%, 1.50%, and 1.33% over F-PointNet+Transformer for the three detection levels, respectively. Fig 9(b) shows the comparison of the validation results under the threshold of IoUBEV = 0.7. The PVRCNN+Transformer method proposed in the study is still superior. This indicates that the optimization of PPC distribution using Transformer network has validity and reliability. On this basis, the study further qualitatively analyzes the proposed method using images from the KITTI dataset. Fig 10 displays the particular outcomes.

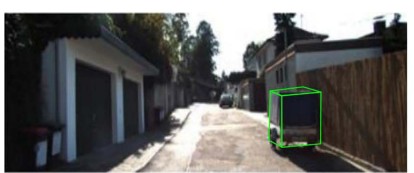

## (a) Original image

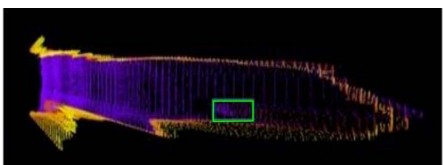

## (b) Pre-optimisation

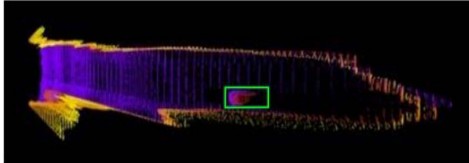

## (c) Post-optimisation

**Fig 10. Pseudo-point cloud optimization qualitative analysis results.**

In Fig 10, after the Transformer optimization, the shape of the vehicle contour is more obvious and the PC features are more easily captured. This indicates that the study proposes that the optimization of PPC using Transformer network can effectively improve the target PC features, thus improving the detection efficiency of the 3D target detector.

### 3.3. Validation of monocular vision measurement method based on pseudo point cloud distribution optimization

The efficiency of the suggested Transformer model to optimize the PPC distribution is confirmed by the earlier validation findings. To further confirm the effectiveness of the proposed monocular vision measurement method, the study introduces the popular algorithms in the current KITTI test set for quantitative analysis and comparison. These mainly include depth regression model (DRM), Did-m3d, feature aggregation strategy (FAS) based 3D target detection and stereo multi-granularity 3D (SGM3D) [35–38]. When IoU3D=0.7, the evaluation results obtained from 10 trials of different detection methods in three detection levels are shown in Fig 11.

Fig 11 shows the validation results of the different methods at a single level of the KITTI test set for IoU3D = 0.7. In Fig 11(a), the results of 10 experiments of the proposed method under study are at the highest value. With the increase of the dataset level, the test results of all the five methods in Fig 11(b) and Fig 11(c) moderate and difficult levels are significantly decreased. Meanwhile, the study further compares the results of 10 tests of different methods at IoUBEV = 0.7. This is specifically shown in Fig 12.

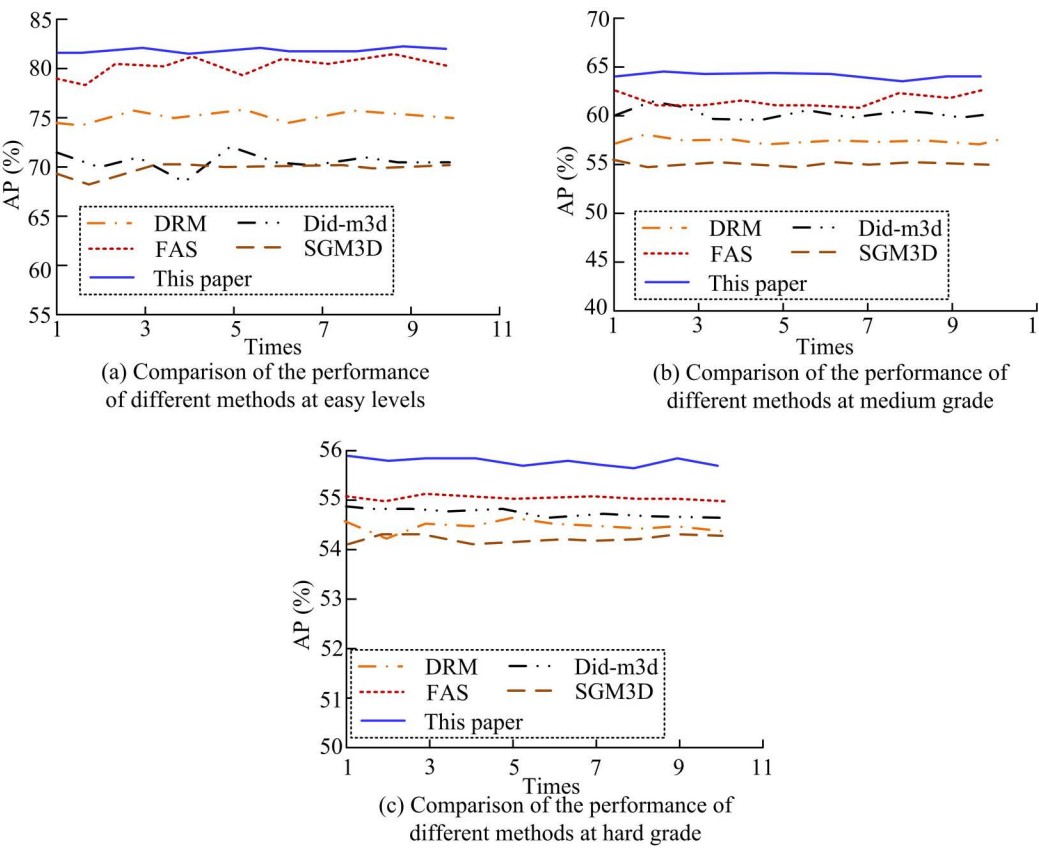

Fig 11. Performance comparison of different methods for IoU3D = 0.7.

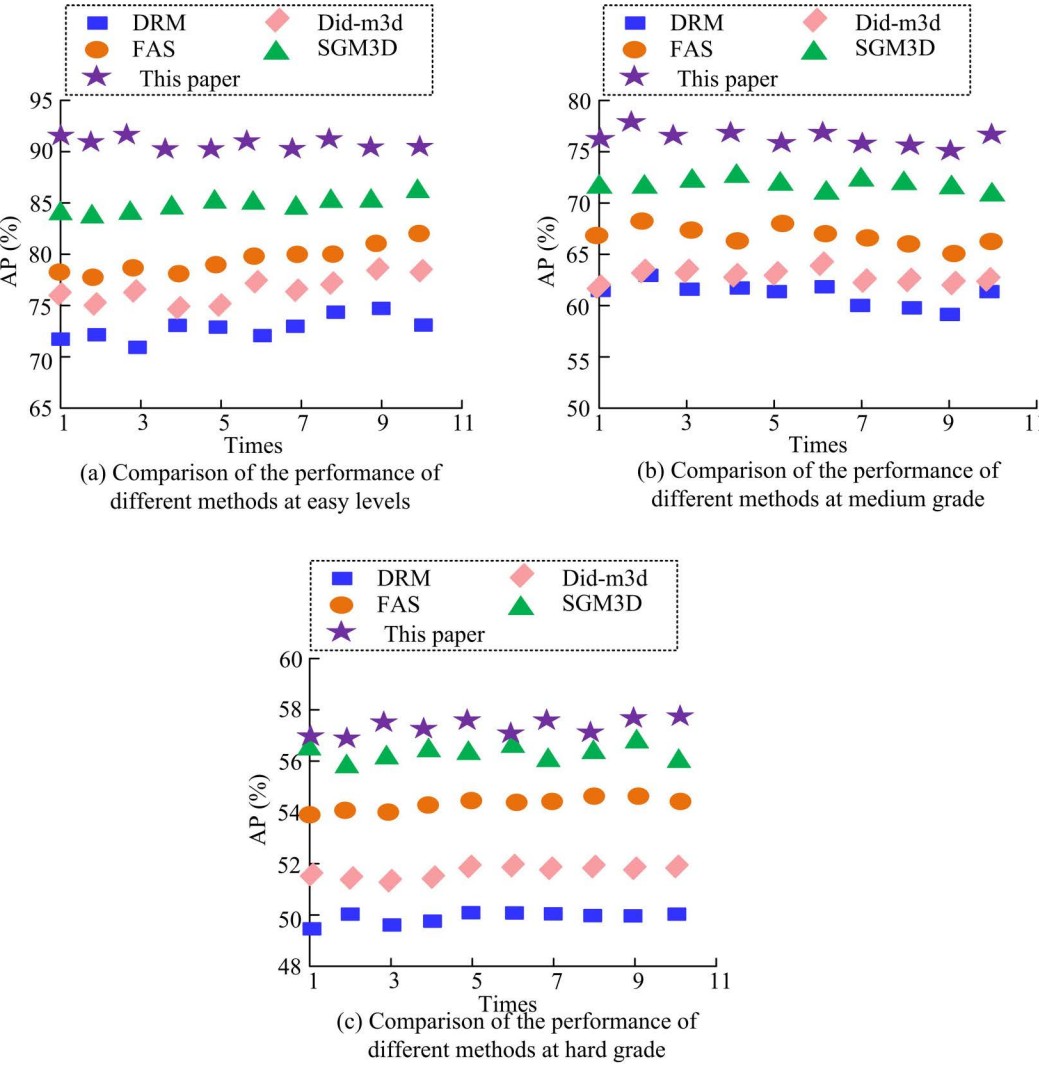

**Fig 12. Performance comparison of different methods for IoUBEV = 0.7.**

Comparing the AP values of 10 measurements of different methods in Fig 12(a), it can be noticed that at IoUBEV = 0.7, the AP value of the study's proposed measurement method is about 91.03%. Combining the detection results in Fig 12(b) and Fig 12(c), the validity of the study's proposed method can be further confirmed. Table 2 displays the AP values derived from the five approaches that is tested ten times.

In Table 2, when IoU3D = 0.7, the study of the proposed method increases the AP value under simple setting by 10.71% on average compared to other methods. Whereas, the study of the proposed method under difficult setting has increased by 2.51%, 1.92%, 1.47%, and 3.03% than the other methods, respectively. However, at IoUBEV = 0.7, the study of the proposed method increases by an average of 16.41%, 14.76%, and 7.85% under different difficulty settings than the other methods, respectively. This indicates that after optimizing the depth of the foreground target and then using Transformer for PPCDO, more accurate and effective depth information can be obtained, which leads to more desirable 3D target detection results. It can be inferred that as the complexity of the scene increases, the efficacy of the proposed model diminishes. This phenomenon may be attributed to the fact that the target object in the simple scene is more discernible

**Table 2. Comparison of the performance of different methods.**

| Method | IoU3D = 0.7 (%) | | | IoUBEV = 0.7 (%) | | |
|---|---|---|---|---|---|---|
| | Easy | Medium | Hard | Easy | Medium | Hard |
| DRM | 75.96 | 57.28 | 54.48 | 72.33 | 62.56 | 49.69 |
| Did-m3d | 71.85 | 61.20 | 54.80 | 75.98 | 64.35 | 51.96 |
| FAS | 80.56 | 62.59 | 55.04 | 79.45 | 66.78 | 54.38 |
| SGM3D | 69.95 | 55.07 | 54.21 | 85.02 | 71.97 | 56.45 |
| This study | 82.57 | 64.75 | 55.85 | 91.03 | 76.22 | 57.29 |

and the background interference is minimal, thereby enabling the model to extract the target features with greater precision. However, in moderate and difficult scenes, the target object has severe occlusion, illumination change, or complex background, which poses a great challenge to the model in feature extraction and depth estimation. The precision, recall, F1 score and false positive rate (FPR) of DRM, Did-m3d, FAS, SGM3D, and the methods proposed in this study are shown in Table 3.

In Table 3, the accuracy, recall, and F1 score of the proposed method are superior to other methods in different difficulty scenes. Moreover, the detection performance is obviously superior in simple and medium difficulty scenes. This shows that this method has high accuracy and robustness in dealing with 3D target detection tasks of varying difficulty. Comparing the FPR of the five methods, the FPR of DRM is the highest at 15%. This indicates that there is a 15% chance that the DRM method will incorrectly predict all negative samples as positive samples. This may indicate that the DRM method faces challenges in accurately distinguishing between positive and negative samples, especially in complex or challenging scenarios. The DRM method relies primarily on a deep regression model and is sensitive to occlusion, illumination changes, and background clutter in complex scenes. In medium and high complexity scenes, the target object may be partially occluded or appear close to the background depth. This can hinder the model's ability to accurately distinguish the target from the background, thereby reducing the detection accuracy. The feature fusion method of DID-M3D is simple, and it can't effectively deal with the deep overlap between the backgrounds of the target area. The FAS method ignores the comprehensive consideration of multi-scale features and local features when dealing with complex scenes. It is difficult to effectively deal with local occlusion and background interference of targets. The FPR of the proposed method is 8.90%, which is the lowest among all methods, indicating that it has the best performance in controlling false positives.

From the above results, it can be seen that the monocular vision measurement method based on the optimisation of PPC allocation shows obvious advantages in different difficulty scenarios. Compared with DRM, Did-m3d, FAS, and SGM3D, the proposed method has higher AP values at all difficulty levels. The greatest improvement in detection performance was achieved in the "difficult" scenario. This indicates that the depth estimation and target detection models optimised for PPC allocation can provide higher accuracy and robustness when dealing with complex backgrounds, target

**Table 3. Comparison of the testing result of different methods.**

| Method | Precision (%) | | | Recall (%) | | | F1 score (%) | | | Average FPR (%) |
|---|---|---|---|---|---|---|---|---|---|---|
| | Easy | Medium | Hard | Easy | Medium | Hard | Easy | Medium | Hard | |
| DRM | 85.02 | 75.58 | 60.24 | 82.38 | 70.48 | 55.57 | 83.68 | 72.94 | 57.81 | 15.00 |
| Did-m3d | 88.97 | 77.65 | 64.32 | 85.56 | 73.24 | 59.98 | 87.23 | 75.38 | 62.07 | 12.34 |
| FAS | 90.01 | 80.27 | 68.88 | 86.78 | 76.56 | 65.90 | 88.37 | 78.37 | 67.36 | 10.21 |
| SGM3D | 87.80 | 79.85 | 66.34 | 86.36 | 76.25 | 62.30 | 87.07 | 78.01 | 64.26 | 14.56 |
| This study | 91.15 | 83.26 | 70.97 | 89.55 | 81.98 | 69.04 | 90.34 | 82.62 | 69.99 | 8.90 |

occlusion and illumination changes. Furthermore, the proposed method effectively reduces false predictions and improves detection efficiency, underscoring its strong application potential in real-world monocular vision measurement tasks.

### 3.4. Wire pole tower external breakage hidden danger example validation

The study uses the inspection data from China Guangdong Power Grid Company as the training set and test set in order to further validate the applicability of the suggested strategy. Its tower categories mainly include V-tower and T-tower. After manually calibrating the positions of the tower PC and PL PC, the collected PCD is expanded using operations such as translation, cropping, mirroring and rotation. Finally, 18840651 experimental data points is obtained. The study categorizes the data into 3 groups of data based on the topography of the area where the data is collected and the type of tower. Experimental data group 1 is mainly dry and T-shaped towers. Its width is 247.15m, length is 6666.89m, and the number of PCs is 15479342. Experimental data group 2 is mainly ram's horn tower. Its width is 64.38m, length is 1606.63m, and the PCs is 1165449. The experimental data 3 groups are mainly dry and T-shaped towers. Its width is 454.06m, length is 3973.31m, and the number of PCs is 2195.860. At the same time, the training set of the previously mentioned KITTI dataset is utilized to train the proposed method model. The study initially evaluates the effectiveness of the suggested approach for the extraction of transmission towers and PLs for three sets of data using the experimental data conditions mentioned above. The specific results are shown in Fig 13.

The results of the study's suggested method for extracting PL points for each of the three data groups are compared with the points that is manually calibrated in Fig 13(a). The suggested method's extraction accuracy in experimental data group 1 is 95.30%. For experimental data groups 2 and 3, the extraction accuracy is 96.77% and 97.81%, respectively. The lower extraction accuracy in experimental data group 1 may be due to the fact that the topography of the region is more complex, with greater undulations and vertical shading. In contrast, the topography of the experimental area in data group 2 and data group 3 is relatively flatter, with less occlusion, higher PC density and less background interference. Therefore, the PCD of the groups 2 and 3 are more complete and the extraction accuracy is higher, which makes the PC extraction method more effective in identifying and extracting targets. In Fig 13(b), the extraction accuracy of the proposed method of the study is 91.95%, 95.83%, and 93.90% in the extraction results of transmission pole tower points for the three sets of test data, respectively. This indicates that the measurement method proposed by the study is able to extract the PCD of transmission lines and transmission towers better, especially for flat terrain. The comparison results of DRM, Did-m3d, FAS, SGM3D and the methods proposed in this study are shown in Table 4.

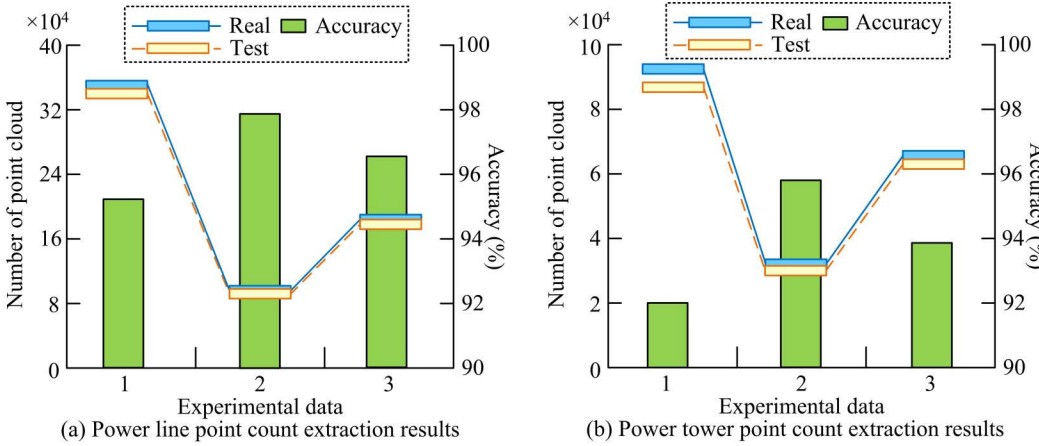

(a) Power line point count extraction results
(b) Power tower point count extraction results

**Fig 13. Extraction results of the proposed method for transmission towers and power lines in three sets of experimental data.**

In Table 4, the accuracy of the proposed method is higher than other methods in all datasets, with an average accuracy of 95.23%. The preprocessing time of the proposed method is 0.48 s/frame, which is the lowest among all the methods. This shows that this method is more efficient in the data preprocessing stage. This may be because the processing flow of PC data is optimized. Meanwhile, it also shows that the proposed method can extract and analyze the PCD of transmission towers more efficiently in practical application. Therefore, the study further conducted a qualitative analysis of examples. The Guangzhou Bureau's 110kV Helongzhong line is used as an example to demonstrate the efficacy of the suggested technique for determining the spatial separation between transmission towers. It is specifically shown in Fig 14.

Fig 14 shows that the distance measurement result of 110kV Hap long center line by the proposed method of the study is 14.87m, while the actual distance of the transmission tower is 15.36m. The accuracy of the measurement is 96.81%, and the error is 0.49m. This shows that the real-time measurement method proposed by the study can effectively measure the distance of the transmission tower. Moreover, after optimizing the PPC distribution using Transformer, the detection target is more easily captured by the 3D detector.

## 4. Discussion

The real-time spatial distance measurement method for transmission tower external breakage hazards proposed in this study demonstrated significant improvements over other methods proposed by scholars, such as DRM, Did-m3d, FAS, and SGM3D. On average, it increased by 16.41%, 14.76%, and 7.85% in different difficulty levels, respectively. The precision rates in three scenarios reached as high as 91.15%, 83.26%, and 70.97%. The measurement method proposed in this study also showed superior performance in the validation with real data of transmission tower poles. This method adopted the Transformer network as the core of PPCDO. The Transformer network could effectively capture global

**Table 4. Comparison of actual data testing by different methods.**

| Method | Accuracy (%) | | | Time overhead (s/frame) | |
|---|---|---|---|---|---|
| | Data set 1 | Data set 2 | Data set 3 | Data preprocessing | Model reasoning |
| DRM | 90.21 | 92.17 | 93.55 | 2.15 | 1.47 |
| Did-m3d | 91.53 | 93.01 | 94.05 | 1.23 | 0.94 |
| FAS | 92.00 | 93.54 | 94.52 | 0.91 | 0.76 |
| SGM3D | 91.87 | 93.22 | 94.26 | 1.34 | 1.02 |
| This study | 93.63 | 96.82 | 95.24 | 0.48 | 0.32 |

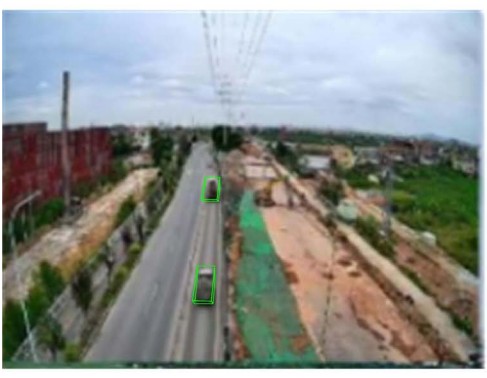

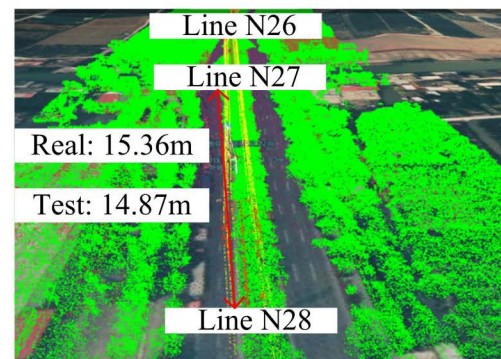

(a) Example site image  (b) Distance measurement results

**Fig 14. Effectiveness of measurement of spatial distance of transmission towers.**

dependencies through self-attention mechanisms, which was crucial for improving the global consistency and local detail accuracy of PPC. To address the inaccuracy of depth estimation for foreground targets in monocular vision, this study proposed a depth optimization method based on 2D instance segmentation and geometric constraints. By optimizing the depth of foreground targets, it improved the accuracy of foreground targets in PPC, thereby improving the overall detection efficiency.

It is noteworthy that the study revealed that a low learning rate facilitates the model's stable convergence during the initial training phase. However, with the advancement of training, an appropriate increase in the learning rate could expedite convergence and enhance the final detection accuracy. A larger batch size could facilitate more stable gradient estimation. However, it also entailed an increase in memory consumption. Through experiments, it was found that moderate batch size could not only ensure the training efficiency of the model, but also did not cause memory overflow. The adjustment of the super-parameters had a crucial influence on the model performance. In the follow-up work, the optimization and adjustment of model super-parameters will be strengthened to improve the model performance.

## 5. Conclusion

The study aimed to improve the detection efficiency of external damage hazards to transmission towers and the accuracy of 3D target detection. By proposing a real-time spatial distance measurement method based on monocular vision, the research not only achieved efficient monitoring of external hazards to transmission towers, but also significantly improved the feature representation, resolution, and depth information of target PC features. In this study, a Transformer network was used to optimize the distribution of PPC, and a foreground target depth optimization method based on a 2D detector was proposed. The training effect of the model was further improved by UL. The validation on the KITTI dataset showed that the optimized PPC distribution could more accurately capture target point cloud features, thereby improving the detection efficiency of the 3D target detector. Fig 9 reveals the superiority of the proposed method of the study in combination with PointNet++. Compared to F-PointNet+Transformer, the AP values of the proposed method in this study are improved by 1.35%, 1.50% and 1.33% at the three detection levels, respectively. Combined with the performance comparison test of different methods in Table 2, it can be seen that the average AP values of the proposed method of the study are improved by 16.41%, 14.76% and 7.85% compared to the other methods for different difficulty settings of IoUBEV = 0.7, respectively. The above results show that by optimising the depth information of foreground targets, the accuracy of foreground targets in PPC can be improved, thus increasing the overall detection efficiency. In the actual validation with transmission tower data, the method proposed in this study achieved an average accuracy rate of 96.56% and 93.89% in extracting PL points and transmission tower points, respectively, further proving the effectiveness of the method.

Nevertheless, the efficacy of the method presented in this study is contingent upon the accessibility of training data and the necessity for manual annotation. In future work, intelligent algorithms will be considered for the design of semi-supervised or self-supervised detection algorithms, with the aim of enhancing their application value in practical power engineering. Secondly, although the study has yielded promising results on the KITTI dataset and actual transmission tower data, its performance on other types of datasets remains to be validated, which will be addressed in future work.

## Supporting information

**S1 File. Minimal data set.**
(DOCX)

## Author contributions

**Conceptualization:** Ruchao Liao.

**Data curation:** Ruchao Liao, Duanjiao Li.

**Formal analysis:** Changyu Li.

**Funding acquisition:** Wenxing Sun, Gao Liu.

**Investigation:** Duanjiao Li.

**Methodology:** Wenxing Sun.

**Project administration:** Changyu Li, Gao Liu.

**Resources:** Duanjiao Li, Changyu Li.

**Software:** Cong Wang.

**Supervision:** Wenxing Sun, Cong Wang.

**Validation:** Gao Liu, Cong Wang.

**Writing – original draft:** Ruchao Liao.

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
