## [Decision Letter · Decision Letter 0]

Dear Dr. Liao,

Thank you for submitting your manuscript to PLOS ONE. After careful consideration, we feel that it has merit but does not fully meet PLOS ONE’s publication criteria as it currently stands. Therefore, we invite you to submit a revised version of the manuscript that addresses the points raised during the review process.

We look forward to receiving your revised manuscript.

Kind regards,

Akhtar Rasool, Ph.D.

Academic Editor

PLOS ONE

Additional Editor Comments:

Please address each comment of the reviewers and also improve the article or expand it, as and where needed. The comments of the reviewers are appended below.

Reviewers' comments:

Reviewer's Responses to Questions

**Comments to the Author**

1. Is the manuscript technically sound, and do the data support the conclusions?

Reviewer #1: Partly

Reviewer #2: Yes

2. Has the statistical analysis been performed appropriately and rigorously?

Reviewer #1: No

Reviewer #2: Yes

3. Have the authors made all data underlying the findings in their manuscript fully available?

Reviewer #1: No

Reviewer #2: No

4. Is the manuscript presented in an intelligible fashion and written in standard English?

Reviewer #1: Yes

Reviewer #2: No

Reviewer #1: This paper proposes a pseudo-point cloud distribution optimization method based on Transformer networks, which demonstrates excellent performance in improving target point cloud features and enhancing the efficiency of 3D object detection. Additionally, it introduces technological innovations such as the Geometric Constraint Depth Refinement (GCDR) module, further boosting measurement accuracy and stability.

However, I have some suggestions, as follows.

1) The article merely elaborates on the importance of safety monitoring for transmission towers and lacks a detailed comparison and analysis of existing technologies for predicting external damage risks to transmission towers. The authors are advised to include a comprehensive review of existing technologies in the introduction and clearly identify the shortcomings of current technologies.

2) The submission contains some formulas and charts that lack sufficient explanation and description. For instance, Figures 4 and 6 lack necessary textual explanations and annotations are not detailed enough. The authors should provide more detailed annotations and explanations for these formulas and charts, including more detailed labeling of specific steps and data flow between modules. Formulas (2), (3), and (4) lack specific explanations and should be briefly explained before or after the formulas to make the mathematical equations easier to understand.

3) Although there is some technical description of the pseudo-point cloud distribution optimization method based on Transformer networks and the depth optimization method based on 2D instance segmentation and geometric constraints, not much detailed description is provided. The authors are advised to add more technical details, including specific steps of the algorithm, parameter settings, and experimental environments.

4) The experimental results presented in the article lack sufficient explanation and analysis of the experimental data. For example, for experimental results in different difficulty scenarios, the authors simply list accuracy, recall, and F1 scores without in-depth discussion of the experimental results. The authors should increase the detailed explanation and analysis of the experimental data, including the source of the experimental data, a detailed description of the experimental process, and an assessment of the reliability and validity of the experimental results.

Therefore, the authors are advised to make major revisions before resubmitting.

Reviewer #2: The authors presented a real-time spatial distance measurement approach using monocular vision together with a Transformer network, enabling the prediction of external breakage hazards in transmission towers. They demonstrated the effectiveness of their method on the KITTI dataset as well as actual tower data, and compared it with other methods.

Strengths:

Their method is well-explained, and the use of monocular vision and Transformer networks contributes to the power engineering. The experiments, figures and tables are clear and support their findings.

Weakness:

1. The authors only compared different method on the KITTI dataset, it would be important to use different methods on the actual tower data too.

2. Since this is a “real-time” measurement method, maybe I missed it, I am wondering how much time it will take to run a trained model on the test data. It would be also interesting to compare the running time for different methods.

Language and Formatting:

There are many grammatical errors in the manuscript. The authors should proofread it very carefully. For example, in the Abstract, “The innovation of the study liedin the optimization of the pseudo point cloud distribution using the Transformer network” should be corrected, “ … lied in …”. In the Conclusion, “This result indicatedthat optimizing the depth information of foreground targets couldsignificantly improve the accuracy of foreground targets in PPC” also requires update.

**Do you want your identity to be public for this peer review?** For information about this choice, including consent withdrawal, please see our Privacy Policy

Reviewer #1: No

Reviewer #2: No

---

## [Author Response · Author response to Decision Letter 1]

21 Mar 2025

The manuscript content has been revised.

---

## [Decision Letter · Decision Letter 1]

Dear Dr. Liao,

Thank you for submitting your manuscript to PLOS ONE. After careful consideration, we feel that it has merit but does not fully meet PLOS ONE’s publication criteria as it currently stands. Therefore, we invite you to submit a revised version of the manuscript that addresses the points raised during the review process.

We look forward to receiving your revised manuscript.

Kind regards,

Akhtar Rasool, Ph.D.

Academic Editor

PLOS ONE

**Journal Requirements:**

**Additional Editor Comments:**

The reviewers have acknowledged the improvements. However, they still have some minor concerns which need to be addressed so it is suggested to address point-wise all the remaining concerns. Please make improvements or modifications in the manuscript where indicated or is required as per the reviewers' comments. 

Reviewers' comments:

Reviewer's Responses to Questions

**Comments to the Author**

Reviewer #1: All comments have been addressed

Reviewer #2: All comments have been addressed

2. Is the manuscript technically sound, and do the data support the conclusions?

Reviewer #1: Partly

Reviewer #2: Yes

3. Has the statistical analysis been performed appropriately and rigorously?

Reviewer #1: Yes

Reviewer #2: Yes

4. Have the authors made all data underlying the findings in their manuscript fully available?

Reviewer #1: Yes

Reviewer #2: Yes

5. Is the manuscript presented in an intelligible fashion and written in standard English?

Reviewer #1: Yes

Reviewer #2: Yes

**Reviewer #1: ** This paper proposes an optimization for pseudo point cloud (PPC) power distribution using transformer-based approaches. Building upon this foundation, it integrates lidar-based Simultaneous Localization and Mapping (SLAM) with deep learning to enhance the collaborative robotic surveying accuracy. The experimental design is well-conceived, and the conclusions offer valuable reference significance for intelligent inspection of power transmission lines and the development of visual technologies in electrical engineering. After review, I consider the paper demonstrates clear logical structure and coherent reasoning that meets publication standards. However, the following issues require further refinement before final acceptance:

Revision Suggestions:

1) While the technical details of the proposed transformer-based pseudo point cloud (PPC) distribution optimization method and the depth refinement approach integrating 2D instance segmentation with geometric constraints are well-improved, the authors are advised to enhance readability by adding intuitive explanations or citing relevant literature following this section.

2) Section 3.3, which validates the monocular visual measurement method based on PPC distribution optimization, is sufficiently comprehensive. However, a concise summary of the data comparison results in this section is recommended to further strengthen its structural coherence.

3) The explanation for Figure 13 is overly brief, which may hinder readers’ intuitive understanding. It is suggested that the authors include a brief explanatory summary beneath the figure to improve clarity.

4) In the concluding section, the authors should incorporate specific references to figures or tables (e.g., “Figure 2 clearly demonstrates…” or “The data in Table 3 proves…”). Integrating such references with textual descriptions will enrich the paper’s structural integrity and analytical depth.

Overall Evaluation:

The aforementioned issues pertain to minor refinements in details and do not affect the core conclusions of the paper. If the authors can address these points, the manuscript is worthy of acceptance.

**Reviewer #2: ** The authors have addressed my previous comments. Thank you for that!

I have one minor suggestion: In Figure 8, the AP values for the unoptimized method appear to be increasing. Could the authors train the model until performance plateaus?

**Do you want your identity to be public for this peer review?** For information about this choice, including consent withdrawal, please see our Privacy Policy

Reviewer #1: No

Reviewer #2: No

---

## [Author Response · Author response to Decision Letter 2]

13 May 2025

The manuscript has improved the quality.

---

## [Decision Letter · Decision Letter 2]

Real-time Measurement of Spatial Distance to External Breakage Hazards of Transmission Pole Tower Based on Monocular Vision

PONE-D-24-48904R2

Dear Dr. Liao,

We’re pleased to inform you that your manuscript has been judged scientifically suitable for publication and will be formally accepted for publication once it meets all outstanding technical requirements.

Kind regards,

Akhtar Rasool, Ph.D.

Academic Editor

PLOS ONE

Additional Editor Comments (optional):

Congratulations

Reviewers' comments:

Reviewer's Responses to Questions

**Comments to the Author**

Reviewer #1: All comments have been addressed

Reviewer #2: All comments have been addressed

2. Is the manuscript technically sound, and do the data support the conclusions?

Reviewer #1: Yes

Reviewer #2: Yes

3. Has the statistical analysis been performed appropriately and rigorously?

Reviewer #1: Yes

Reviewer #2: Yes

4. Have the authors made all data underlying the findings in their manuscript fully available?

Reviewer #1: Yes

Reviewer #2: Yes

5. Is the manuscript presented in an intelligible fashion and written in standard English?

Reviewer #1: Yes

Reviewer #2: Yes

Reviewer #1: The authors have made necessary revisions, for example, all of the format problems have been revised, and so are the clarifications.

Reviewer #2: (No Response)

**Do you want your identity to be public for this peer review?** For information about this choice, including consent withdrawal, please see our Privacy Policy

Reviewer #1: No

Reviewer #2: No

---

## [Editor Report · Acceptance letter]

PONE-D-24-48904R2

PLOS ONE

Dear Dr. Liao,

I'm pleased to inform you that your manuscript has been deemed suitable for publication in PLOS ONE. Congratulations! Your manuscript is now being handed over to our production team.

Kind regards,

on behalf of

Dr. Akhtar Rasool

Academic Editor

PLOS ONE